# Retbindin: A riboflavin Binding Protein, Is Critical for Photoreceptor Homeostasis and Survival in Models of Retinal Degeneration

**DOI:** 10.3390/ijms21218083

**Published:** 2020-10-29

**Authors:** Ayse M. Genc, Mustafa S. Makia, Tirthankar Sinha, Shannon M. Conley, Muayyad R. Al-Ubaidi, Muna I. Naash

**Affiliations:** 1Department of Biomedical Engineering, University of Houston, Houston, TX 77204, USA; amgenc@Central.UH.EDU (A.M.G.); msmakia@Central.UH.EDU (M.S.M.); Tirthankar.Sinha@uth.tmc.edu (T.S.); 2Department of Cell Biology, University of Oklahoma Health Sciences Center, Oklahoma City, OK 73104, USA; Shannon-Conley@ouhsc.edu; 3Oklahoma Center for Neurosciences, University of Oklahoma Health Sciences Center, Oklahoma City, OK 73104, USA; 4College of Optometry, University of Houston, Houston, TX 77204, USA; 5Department of Biology and Biochemistry, University of Houston, TX 77204, USA

**Keywords:** *Prph2*, retinal degeneration, inherited retinal degeneration, riboflavin, metabolism, flavoproteins, rhodopsin

## Abstract

The large number of inherited retinal disease genes (IRD), including the photopigment rhodopsin and the photoreceptor outer segment (OS) structural component peripherin 2 (PRPH2), has prompted interest in identifying common cellular mechanisms involved in degeneration. Although metabolic dysregulation has been shown to play an important role in the progression of the disease etiology, identifying a common regulator that can preserve the metabolic ecosystem is needed for future development of neuroprotective treatments. Here, we investigated whether retbindin (RTBDN), a rod-specific protein with riboflavin binding capability, and a regulator of riboflavin-derived cofactors flavin mononucleotide (FMN) and flavin adenine dinucleotide (FAD), is protective to the retina in different IRD models; one carrying the P23H mutation in rhodopsin (which causes retinitis pigmentosa) and one carrying the Y141C mutation in *Prph2* (which causes a blended cone-rod dystrophy). RTBDN levels are significantly upregulated in both the rhodopsin (*Rho*)*^P23H/+^* and *Prph2^Y141C/+^* retinas. Rod and cone structural and functional degeneration worsened in models lacking RTBDN. In addition, removing *Rtbdn* worsened other phenotypes, such as fundus flecking. Retinal flavin levels were reduced in *Rho^P23H/+^/Rtbdn^−/−^* and *Prph2^Y141C/+^/Rtbdn^−/−^* retinas. Overall, these findings suggest that RTBDN may play a protective role during retinal degenerations that occur at varying rates and due to varying disease mechanisms.

## 1. Introduction

Inherited retinal degenerations (IRD) comprise a group of widely varying, blinding retinal diseases caused by more than 300 different gene or loci mutations (https://sph.uth.edu/retnet/disease.htm, accessed 19 October 2020). Two of the most commonly mutated genes in patients with IRD are rhodopsin (*RHO*), the rod visual pigment, and peripherin 2 (*PRPH2,* also known as *RDS*), a structural component of the photoreceptor outer segment (OS) rim. Over the past several decades, much work has been done to elucidate the mechanisms of disease for the most common retinal disease genes. However, as the number and diversity of genes responsible for IRD continues to increase, so has interest in identifying mechanisms of promoting photoreceptor homeostasis, which might be beneficial across multiple genes/mutations.

During our prior investigations into photoreceptor metabolic homeostasis, we became interested in a novel retina-specific protein called retbindin (RTBDN) which has homology to a chicken riboflavin binding protein [1,2]. Riboflavin (Vitamin B2) is a vitamin essential for cell growth and function [3]. Intracellularly, it is primarily found in its redox active co-enzymatic forms, flavin adenine dinucleotide (FAD) and flavin mononucleotide (FMN). Flavoproteins play vital roles in a variety of processes, including the citric acid cycle, β-oxidation, amino acid synthesis, protein folding, DNA repair, and normal immune function [4]. Consistent with the high level of metabolic activity in photoreceptors [5], retinal flavin levels are almost 20-fold higher than in blood [6,7].

Given the importance of flavins to retinal metabolic processes, coupled with the lack of knowledge on how the retina acquires and concentrates flavins, we undertook further characterization of the RTBDN protein. It is exclusively expressed by rod photoreceptor cells and is secreted into the inter photoreceptor matrix (IPM), where it distributes predominantly at the junction of the OS and the retinal pigment epithelium (RPE), as well as around the inner segments [1]. Though not a transmembrane protein, RTBDN remains anchored to the plasma membrane via electrostatic forces [1]. In vitro and ex vivo studies showed that RTBDN binds riboflavin, and subsequent investigation revealed that levels of riboflavin and its derivatives are severely reduced in the retinas of *Rtbdn* knockout mice (*Rtbdn^−/−^),* supporting a role for RTBDN in regulating or maintaining steady state levels of retinal flavins [1,8].

The importance of RTBDN for retinal health was highlighted by the observation that *Rtbdn^−/−^* mice exhibit late-onset (~four months of age) decreases in rod and cone function, as well as photoreceptor degeneration [8]. Since metabolic dysregulation is a hallmark of photoreceptor cell death [9] and flavins are critical for metabolism [4], we subsequently evaluated the role of RTBDN in a model of IRD. We found that eliminating *Rtbdn* in the R172W *Prph2* model of macular dystrophy [10] exacerbated *Prph2*-associated retinal disease, suggesting RTBDN may play a protective role in IRD [11]. However, the *Prph2^R172W^* model is associated with early-onset cone degeneration with rod defects being less severe until later, and it was not clear whether eliminating *Rtbdn* would also have an effect on rod-dominant diseases, such as retinitis pigmentosa (RP) or mixed cone-rod dystrophies and pattern dystrophies.

To evaluate whether the protective role of RTBDN applied more broadly to multiple models of IRD with different disease mechanisms, we here evaluated the effects of removing *Rtbdn* in two additional models: the *Rho^P23H/+^* model of RP [12] and the *Prph2^Y141C/+^* model of pattern dystrophy and RP [13]. RP is a form of IRD that occurs in 1 out of 4000 people worldwide [14] and is associated with progressive night blindness, a characteristic fundus spicule phenotype, and ultimately rod degeneration and blindness. The P23H rhodopsin mutation is the most common (~15%) cause of rhodopsin-related RP in the United States [15,16,17], and several models exist to study it [12,18,19,20]. The Y141C mutation in *PRPH2* is associated with a high degree of phenotypic heterogeneity. Some patients are diagnosed with pattern dystrophy, exhibiting macular changes, RPE pigmentation, drusen-like deposits, chorioretinal atrophy, and occasional late-stage choroidal neovascularization, while other patients exhibit a classic RP phenotype [21,22,23,24]. We previously generated a knockin mouse model of the Y141C mutation in which mice develop a cone-rod dystrophy phenotype characterized by severe loss-of-function in rods and cones, retinal degeneration, and fundus phenotypes similar to those seen in pattern dystrophy patients [13]. Here, we find that levels of RTBDN are significantly upregulated in the *Rho^P23H/+^* and *Prph2^Y141C/+^* retinas and that retinal degeneration in these models is exacerbated by removal of *Rtbdn,* suggesting that RTBDN plays a protective role during the degenerative process across multiple different forms of IRD.

## 2. Results

### 2.1. RTBDN Is Upregulated in the Rho^P23H/+^ and Prph2^Y141C/+^ Retina

Here, we utilize two different models of inherited retinal degeneration, the rhodopsin P23H knockin (*Rho^P23H/+^*) model of RP [12,25] and the *Prph2* Y141C knockin (*Prph2^Y141C/+^*) model of cone-rod dystrophy [13] to evaluate whether eliminating retbindin has broad effects on retinal degeneration. The *Rho^P23H/+^* exhibits rapid retinal degeneration, so we initially examined these retinas at postnatal day (P) 15. The degeneration in the *Prph2^Y141C/+^* is slower, so we conducted the earliest experiments at P30. At early timepoints, RTBDN protein levels were significantly upregulated by ~2-fold in both the *Rho^P23H/+^* retina (P15, P30, Figure 1A,B, solid bars) and in the *Prph2^Y141C/+^* retina (P30, Figure 1C,D, solid bars) compared to wild type (WT). At P90, this upregulation was lost; RTBDN levels were similar in *Rho^P23H/+^* and *Prph2^Y141C/+^* compared to WT. However, both of these models exhibit photoreceptor degeneration, so we wondered whether the apparent loss of RTBDN upregulation at P90 was due to degeneration. When RTBDN levels were normalized to the number of remaining photoreceptor cells (counted from histological sections), the pronounced upregulation of RTBDN in our degenerative models was preserved at P90 (Figure 1A,C, white bars).

Previously, we completed extensive biochemical characterization of the subcellular localization of RTBDN protein [1]. In the retina, RTBDN is expressed specifically by rods and is located in the extracellular compartment in the inter-photoreceptor matrix. Though it does not have transmembrane domains, it remains associated with photoreceptor membranes via electrostatic interactions [1]. To determine whether this pattern is altered in degenerative models, retinal extracts from P30 animals were separated into fractions enriched for soluble interphotoreceptor matrix (S-IPM, interphotoreceptor retinoid binding protein (IRBP) is shown as a marker), retinal membranes (PRPH2 is shown as a marker), and retinal cytoplasm (glyceraldehyde 3-phophate dehydrogenase (GAPDH) is shown as a marker). In the WT retina, a small pool of newly synthesized RTBDN is found in the cytoplasm, while the majority is in the membrane fraction (Figure 1E,F). This pattern is not changed in either the *Rho^P23H/+^* (Figure 1E) or in the *Prph2^Y141C/+^* (Figure 1F): the majority of RTBDN remains associated with the membrane fraction. These findings clearly show that in multiple models of retinal degeneration, RTBDN protein levels are dramatically upregulated, but the subcellular localization of RTBDN is not changed.

We next addressed whether the upregulation of RTBDN alters its distribution in the degenerative models. Immunofluorescence localization of RTBDN was performed at P30 (Figure 1G,H). In the wild-type (WT) retina, we have previously shown that RTBDN is extracellular and restricted to two areas, the tip of the OSs at the junction of the OS/RPE and in a pool surrounding the inner segment (IS) [1,8]. Interestingly, in the *Rho^P23H/+^* retina, the pattern of RTBDN labeling was significantly altered. Although the labeling around the IS and at the tip of OS/RPE junction was preserved, large amounts of RTBDN staining were found throughout the OS layer in globular patches rather than the normal diffuse pattern (Figure 1G). Co-labeling with rhodopsin (Figure 1G, green) or peanut agglutinin (PNA, Figure 1G, teal) showed that these large patches of RTBDN did not co-localize with either rod or cone OS, respectively, suggesting RTBDN was accumulating abnormally in the interphotoreceptor matrix around OSs during degeneration in the *Rho^P23H/+^*. In contrast to the pattern in the *Rho^P23H/+^*, although RTBDN labeling was more intense in the *Prph2^Y141C/+^* (due to increased protein levels), the labeling pattern was largely preserved as in WT except for a small increase in expression around OSs (Figure 1H). This altered pattern of RTBDN labeling may be due to known abnormalities in OS structure in the *Prph2^Y141C/+^* and *Rho^P23H/+^* [13,25].

Redistribution of proteins can be a mechanism to adapt or respond to changing environments [26,27,28]. To see whether the immunolabeling changes observed with RTBDN in the *Rho^P23H/+^* retinas occur at different stages of degeneration, we performed immunostaining on *Rho^P23H/+^* retinas at P15, P30, and P90 using antibodies against RTBDN (red, Figure 2) and rhodopsin (green, Figure 2) as an OS marker. The bottom panels in each section represents a 2.5D view in which boxed regions are rotated and fluorescence intensity is shown in the Z-direction. In the WT retina at P15, RTBDN is localized primarily around ISs, largely at their distal edge (Figure 2A). By P30 and later, WT retinas have adopted the adult labeling pattern (around IS and at the tip of OS/RPE junction, Figure 2B,C). However, this distribution is altered from the earliest timepoint examined in *Rho^P23H/+^* retinas. At P15 (the initial stages of degeneration), RTBDN exhibits the abnormal globular pattern of expression in the OS layer (Figure 2A–C, arrows), interspersed with OS rhodopsin labeling, rather than being enriched at the IS/OS junction as RTBDN is at this age in the WT. This globular RTBDN labeling pattern throughout the OS layer is preserved at P30 and P90, during the mid and late stages of degeneration (Figure 2B,C). Interestingly, rhodopsin is mislocalized to the IS and ONL in the *Rho^P23H/+^* at P15 (Figure 2A, arrowheads), consistent with ongoing abnormalities in this model [12,25], though this mislocalization is less prominent as degeneration progresses at P30 and P90 (Figure 2B,C). To further refine our understanding of RTBDN distribution in the mutant retinas, we performed co-labeling with RTBDN and synaptosomal-associated protein, 25kDa (SNAP25) (Figure 2D,E), which nicely outlines the ISs. We found that the abnormal globular labeling seen in the *Rho^P23H/+^* and the *Prph2^Y141C/+^* primarily localizes at the distal tip of the IS (arrows, Figure 2E), rather than around the length of the IS.

### 2.2. Elimination of Rtbdn Exacerbates Structural and Functional Degeneration in the Rho^P23H/+^ and Prph2^Y141C/+^ Retinas

Previously, we found that elimination of *Rtbdn* accelerated degeneration in the *Prph2* R172W model of macular dystrophy [11]. To determine whether this finding was applicable to more than one degenerative model, we assessed retinal structure and function in *Rho^P23H/+^* and *Prph2^Y141C/+^* mice crossed onto the *Rtbdn^−/−^* background. Light microscopic analysis and associated morphometry (Figure 3A,B) confirmed our previous findings that photoreceptor loss does not occur in the *Rtbdn^−/−^* until after P90 [8]. *Rho^P23H/+^* mice exhibit significant outer nuclear layer thinning as early as P15 (blue, Figure 3B, + indicates comparison between WT and *Rho^P23H/+^/Rtbdn^−/−^*, & indicates comparison between WT and *Rho^P23H/+^*, and * indicates comparison between *Rho^P23H/+^* and *Rho^P23H/+^/Rtbdn^−/−^.* One symbol, *p* < 0.05, two symbols, *p* < 0.01, three symbols, *p* < 0.001, and four symbols, *p* < 0.0001 by two-way ANOVA with Sidak’s post-hoc test). This degeneration progresses rapidly, and by P90 only 2–3 rows of photoreceptor nuclei remain. This degeneration is accelerated by removal of *Rtbdn*. At both P15 and P30, photoreceptor degeneration is significantly worse in the *Rho^P23H/+^*/*Rtbdn^−/−^* compared to the *Rho^P23H/+^* (Figure 3B, green). By P90, the degeneration in the *Rho^P23H/+^* has caught up to the *Rho^P23H/+^*/*Rtbdn^−/−^* and ONL thickness is similar in the two models, and severely reduced compared to WT. In addition to photoreceptor loss, we also observe that the OS/IS layer is thinner in the *Rho^P23H/+^*/*Rtbdn^−/−^* compared to both WT and *Rho^P23H/+^* at P15 and P30 (Figure 3A, red bars). As the murine retina is ~95% rods, we assessed cone loss by labeling retinal flat mounts with PNA (which labels the cone matrix sheath, Figure 3C). At P30, no cone loss was detected in the *Rho^P23H/+^* (Figure 3D), consistent with P23H as a model of retinitis pigmentosa where cone loss occurs later, a secondary phenotype following primary rod death. However, we did observe a significant reduction in cone number between the *Rho^P23H/+^*/*Rtbdn^−/−^* compared to the *Rho^P23H/+^*.

Overall, *Prph2^Y141C/+^* retinas exhibited a similar phenomenon, wherein *Rtbdn* elimination worsens degeneration. Photoreceptor loss is slower in the *Prph2^Y141C/+^* than in the *Rho^P23H/+^*, and at P30 only slight degeneration was observed in the *Prph2^Y141C/+^* (Figure 4A,C). However, by P90, significant degeneration was observed in the *Prph2^Y141C/+^* compared to WT, and degeneration was significantly worse in the *Prph2^Y141C/+^/Rtbdn^−/−^* than the *Prph2^Y141C/+^* (Figure 4C, + indicates comparison between WT and *Prph2^Y141C/+^/Rtbdn^−/−^*, & indicates comparison between WT and *Prph2^Y141C/+^*, and * indicates comparison between *Prph2^Y141C/+^* and *Prph2^Y141C/+^/Rtbdn^−/−^.* One symbol, *p* < 0.05, two symbols, *p* < 0.01, three symbols, *p* < 0.001, and four symbols, *p* < 0.0001 by two-way ANOVA with Sidak’s post-hoc test). The OS/IS layer also exhibited thinning in the *Prph2^Y141C/+^* at P30 and P90 (Figure 4A, red bars), another phenotype which was worsened in the *Prph2^Y141C/+^/Rtbdn^−/−^*. To evaluate whether *Rtbdn* elimination affects cone degeneration in the *Prph2^Y141C/+^*, retinal cross-sections were labeled with PNA at P90, and cones were counted (Figure 4B,D). At P90, cone loss is modest but not statistically significant in both the *Prph2^Y141C/+^* and the *Prph2^Y141C/+^/Rtbdn^−/−^* compared to WT, and the elimination of *Rtbdn* did not accelerate or worsen this phenotype at early timepoints (Figure 4B,D).

To further explore the structure of OS and IS in our degenerative models, we performed transmission electron microscopy on retinal thin sections at P30. Consistent with previous findings, the *Rtbdn^−/−^* exhibited no significant ultrastructural changes at this age (Figure 5A, second panel) [8]. Here, we find a similar phenotype in the *Rho^P23H/+^* retina as previously reported [12,25] to have shorter, disorganized OSs with perpendicularly aligned discs (Figure 5A, third panel). Elimination of *Rtbdn* in the *Rho^P23H/+^* retina led to photoreceptors with very short and sparse OSs (Figure 5A, fourth panel, white arrows). Examination at higher magnification highlights further defects in OS structure (Figure 5B). Misoriented discs are easily visible in the *Rho^P23H/+^* (Figure 5B, white arrowheads in the third panel). In the *Rho^P23H/+^/Rtbdn^−/−^*, OSs were severely malformed, frequently exhibiting unflattened discs and significant accumulation of abnormal membranous material (Figure 5B, fourth panel, black arrows and black arrowheads, respectively).

*Prph2^Y141C/+^* retinas also exhibited ultrastructural defects in the outer retina. *Prph2^Y141C/+^* OSs were shorter, wider, and sparser than in the WT (Figure 5A, fifth panel). These phenotypes were exacerbated in the *Prph2^Y141C/+^/Rtbdn^−/−^*: OSs were very short and round with relatively few OSs present (Figure 5A, white arrows, sixth panel). Examination at higher magnification highlighted modest accumulation of abnormal membranous material at the base of the OS in the *Prph2^Y141C/+^* (Figure 5B, fifth panel, black arrowheads), a finding that was worsened in the *Prph2^Y141C/+^/Rtbdn^−/−^* (Figure 5B, sixth panel, arrowheads). *Prph2^Y141C/+^/Rtbdn^−/−^* OSs also exhibited whorl formation (Figure 5B, sixth panel, “W”).

To determine whether these structural changes had functional correlates, we performed full-field electroretinography (ERG) on the *Rho^P23H/+^/Rtbdn^−/−^* and *Prph2^Y141C/+^/Rtbdn^−/−^* mice (and controls) at various timepoints (representative ERG traces at P30 are shown in Figure 6A,D). Consistent with our previous studies showing that *Rtbdn^−/−^* do not exhibit ERG defects until P120 [8], we observed no difference in rod (scotopic) or cone (photopic) responses in WT versus *Rtbdn^−/−^* at P15, P30, P60, or P90 (Figure 6C–F). The *Rho^P23H/+^* exhibits significantly reduced scotopic a-wave responses (compared to WT) as early as P15, which progressively worsen with age (Figure 6B). Cone responses in the *Rho^P23H/+^* were not significantly reduced until P60 (Figure 6C). Eliminating *Rtbdn* in the *Rho^P23H/+^* had no effect on rod function at P15, but, by P30, a significant decline in rod function became apparent (Figure 6B,C). At P60 and P90, loss of scotopic ERG in the *Rho^P23H/+^* caught up to that in the *Rho^P23H/+^/Rtbdn^−/−^*, and there was no significant difference between the groups, with both being reduced by 80% compared to WT. At P15, cone responses were not different in the *Rho^P23H/+^/Rtbdn^−/−^* compared to the *Rho^P23H/+^* or compared to WT, but, at both P30 and P60, cone function was significantly worse in the *Rho^P23H/+^/Rtbdn^−/−^* than in the *Rho^P23H/+^* (Figure 6C). Again, by P90, loss of cone function was similar in the *Rho^P23H/+^* and *Rho^P23H/+^/Rtbdn^−/−^*, likely reflecting the advanced state of degeneration in these animals.

The pattern was slightly different in the *Prph2^Y141C/+^* model. *Prph2^Y141C/+^* animals exhibited declines in rod and cone function detectable as early as P30 and worsening over time, consistent with our previous observations (Figure 6E,F). Eliminating *Rtbdn* worsened Y141C-associated decreases in rod function at all-time points examined (Figure 6E), though the difference was not statistically significant at P60. Similarly, cone function was significantly worse in *Prph2^Y141C/+^/Rtbdn^−/−^* than in *Prph2^Y141C/+^* at P30, P60, and P90 (Figure 6F).

### 2.3. Ablation of Rtbdn in the Rho^P23H/+^ and Prph2^Y141C/+^ Leads to Non-Photoreceptor Phenotypes in the Retina

To determine whether eliminating *Rtbdn* in the models leads to other secondary retinal abnormalities, we performed fundus imaging and fluorescein angiography. At both P90 and P150, eyes of *Rtbdn^−/−^* mice exhibited fundus and angiogram appearance similar to WT (Figure 7A–C). At P90, *Rho^P23H/+^* exhibited a normal fundus appearance, but reduced intensity of fluorescein signal, consistent with the previously reported attenuation of their retinal capillary [29]. Fundus images from P90 *Rho^P23H/+^/Rtbdn^−/−^* mice showed minor changes, such as increased fundus speckling (Figure 7A, black arrows in the upper panels), when compared to *Rho^P23H/+^*. Fluorescein angiography of *Rho^P23H/+^/Rtbdn^−/−^* retinas showed that capillary beds were denser than controls and exhibited signs of neovascular tufts (Figure 7A, white arrows in lower panels), similar to what we previously observed in *Prph2^R172W^/Rtbdn^−/−^* retinas [11]. At much later ages, after the retina is almost completely degenerated, large patches where the RPE has been lost are apparent, leading to a highly abnormal angiogram due to some visualization of the underlying choroidal vasculature (Appendix A
Appendix A).

*Prph2^Y141C/+^* mice exhibit pronounced fundus flecking by P180 [13], and we here see early signs of this at P90 with worsened flecking by P150 (Figure 7B,C). These speckles were more numerous and more pronounced in *Prph2^Y141C/+^/Rtbdn^−/−^* retinas (Figure 7B,C, black arrows). Additionally, some of the *Prph2^Y141C/+^/Rtbdn^−/−^* retinas displayed bright patchy spots (Figure 7B,C, white arrowheads) which are similar to those suggested to be drusen-like deposits in the Apo E knock-out mice [30]. Fluorescein angiography showed signs of neovascularization, such as denser capillary beds and some neovascular tufts (white arrows), in *Prph2^Y141C/+^/Rtbdn^−/−^* eyes (Figure 7B,C). These features were absent or attenuated in age matched *Prph2^Y141C/+^* mice.

The fundus speckles are thought to originate in the RPE, so we undertook closer evaluation of the RPE and Bruch’s membrane in our EM tissue from *Prph2^Y141C/+/^Rtbdn^−/−^* retinas. Qualitative examination suggested that autophagosomes/undigested OS material were larger in the *Prph2^Y141C/+^* than in WT and even larger in the *Prph2^Y141C/+^/Rtbdn^−/−^* (Figure 7D, white arrows), suggesting RPE cells may not be functioning normally. We also observed numerous round, electro-lucent, non-membrane bound particles (Figure 7D, white arrowheads) within the Bruch’s membrane in *Rtbdn^−/−^* retinas which were more prevalent in *Prph2^Y141C^^/+^/Rtbdn^−/−^* retinas. These electro-lucent particles scattered throughout Bruch’s membrane may be lipid-associated materials similar to those reported in other models [31,32]. These electro-lucent particles were not observed in the *Prph2^Y141C/+^* (Figure 7D) or *Rho^P23H/+^* retinas (Appendix A), suggesting they are associated with *Rtbdn* deficiency.

### 2.4. Lack of RTBDN in IRD Models Alters Flavin Levels

Since RTBDN binds riboflavin [1] and is upregulated in the *Rho^P23H/+^* and *Prph2^Y141C/+^* retinas, we measured levels of riboflavin and its cofactors, FAD and FMN, in the retina using HPLC. At P15, there were no significant differences in flavin levels between WT, *Rtbdn^−/−^*, *Rho^P23H/+^*, or *Rho^P23H/+^/Rtbdn^−/−^* (Figure 8A–D). At P30, riboflavin, FMN, FAD, and total flavins were reduced in *Rtbdn^−/−^* retinas compared to the WT confirming our previous findings [8]. Despite the elevation of RTBDN levels in the *Rho^P23H/+^*, levels of retinal riboflavin are severely reduced in this model compared to *Rtbdn^−/−^* and WT (Figure 8A). However, FAD and FMN levels were similarly reduced in the *Rho^P23H/+^* and the *Rtbdn^−/−^* (Figure 8B,C), and overall flavin levels were similar in the *Rho^P23H/+^* and *Rtbdn^−/−^* (Figure 8D). Elimination of *Rtbdn* in *Rho^P23H/+^* retina did not alter flavin levels compared to the *Rtbdn^−/−^* alone: levels of riboflavin, FAD, FMN, and total flavins were similarly reduced (compared to WT) in the *Rtbdn^−/−^* and *Rho^P23H^/Rtbdn^−/−^* (Figure 8A–D).

The pattern was slightly different in the *Prph2^Y141C/+^* retinas. Similar to the case with the *Rho^P23H^/Rtbdn^−/−^*, riboflavin, FAD, FMN, and total flavin levels were similar in the *Prph2^Y141C/+^/Rtbdn^−/−^* and the *Rtbdn^−/−^* at P30 and reduced compared to WT (Figure 8F–I). However, riboflavin, FMN, and total flavin levels were significantly higher in *Prph2^Y141C/+^* than in the other groups (Figure 8 F,H,I). In addition, riboflavin made up a much larger portion of the total retinal flavin pool in the *Prph2^Y141C//+^* retina (62%), and a much smaller portion in the *Rho^P23H/+^* (3%), in comparison to WT, *Rtbdn^−/−^*, and the double mutants where riboflavin was 25−35% of the total flavin pool. Once again, to get an idea of how flavin levels were changing in the context of degeneration, we normalized total flavin levels to the remaining number of photoreceptors (Figure 8E,J). Degeneration in the *Prph2^Y141C/+^* models is fairly minor at P30, so flavin levels in these groups are largely unaffected by normalizing to photoreceptor number. In contrast, the degeneration in *Rho^P23H/+^* line and the acceleration of this degeneration in the *Rho^P23H/+^/Rtbdn^−/−^*, the level of flavins per photoreceptor goes up significantly in the *Rho^P23H/+^/Rtbdn^−/−^* (Figure 8E).

Since disturbances in flavin homeostasis are expected to lead to implications for energy production [33], and several retinal disorders are associated with compromised energy metabolism, we measured ATP levels (Appendix A). However, no significant differences in total retinal ATP were detected at P30 in any of the groups.

## 3. Discussion

Here, we utilized two different IRD models to assess whether the effects of removing *Rtbdn* were similar across IRDs associated with different disease mechanisms. Our overall findings are similar to what we previously reported when analyzing the effects of *Rtbdn* ablation in the *Prph2^R172W^* model. Specifically, we find that RTBDN levels are significantly upregulated in both the *Prph2^Y141C/+^* and *Rho^P23H/+^* retina and that degeneration is accelerated in these lines when *Rtbdn* is removed. However, there are also some interesting differences in the response to *Rtbdn* ablation across the three models we have evaluated thus far which are worthy of further discussion.

In all three of our IRD models, removing *Rtbdn* worsened retinal degeneration. However, the magnitude of this effect varied across models inversely with the severity of degeneration in the original model. That is, the degeneration-accelerating effect of removing *Rtbdn* was greatest in the *Prph2^R172W^* (which ordinarily degenerates slowly) [11], and least in the *Rho^P23H/+^* (which degenerates quickly), while the effects on the *Prph2^Y141C/+^* fell in the middle. This is logical: where the original insult causes only slow degeneration, preserving cellular homeostasis (for example by maintaining retinal flavin levels) would be more beneficial than in a quickly degenerating model, such as the *Rho^P23H/+^*, where the initial insult is so severe that photoreceptors die quickly. Similarly, *Rtbdn* removal worsens cone function in *Rho^P23H/+^* (which is normally not affected until later) more than it worsens rod function. However, in addition to the rate of degeneration, it is worth considering the targeted cell-type. The *Prph2^R172W^* mutation targets primarily cones, with rod loss occurring subsequently. The *Prph2^Y141C/+^* represents a blended cone-rod dystrophy model, while the *Rho^P23H/+^* is a rod-dominant disease (and rhodopsin is only expressed in rods) with cone loss occurring later, secondary to rod loss. RTBDN is only expressed in rods, although both rods and cones eventually degenerate in the *Rtbdn^−/−^*. Our findings suggest that the beneficial effects of RTBDN are greatest (i.e., the effects of removing RTBDN are the worst) in models where rod loss is secondary, rather than the primary target, and in models where degeneration is slow/chronic rather than rapid/early-onset.

Further evaluation of the beneficial effects of Rtbdn during degeneration will require evaluation in an overexpressing model. Here, we show that removing Rtbdn accelerates retinal degeneration; however, to develop an effective therapeutic, it will be necessary to demonstrate that overexpression of Rtbdn will provide benefit. While our results suggest that overexpressing Rtbdn would be beneficial, it is not clear what the ceiling would be for such a benefit. That is, we already see that overexpression of Rtbdn occurs during degeneration; would additional overexpression (either prior to the onset of degeneration or during the disease process) provide improvement? Some insight into this may be obtained from the timecourse we see in our Rtbdn knockout lines. As mentioned above, the benefit of Rtbdn appears to be the greatest in slowly degenerating models. This suggests that, in slowly degenerating models, maintaining high Rtbdn levels (beyond what happens endogenously) could provide essential additional pro-survival benefit. This is an exciting avenue for future exploration and one that can only be determined by empirical testing.

Disturbance of metabolic homeostasis is a contributing factor for the degenerative process [34,35,36]. Since RTBDN binds riboflavin and flavins play critical roles in various metabolic processes, upregulation of RTBDN during cell death likely reflects a stress response in the degenerating cells. This may be associated with increased energy demand as part of cell biological processes associated with cell death or survival. However, although RTBDN levels are consistently upregulated across all three IRD models that we have evaluated so far (*Prph2^R172W^*, *Prph2^Y141C/+^*, and *Rho^P23H/+^*), the levels of retinal flavins are not consistent across the three models. In the *Prph2^Y141C/+^*, total flavin levels are increased compared to WT at P30, consistent with increased RTBDN levels. However in the *Rho^P23H/+^*, total retinal flavin levels are severely decreased at P30 (similar to our findings in the *rd1* and *rd10* models [37]), and in the *Prph2^R172W^* where total retinal flavins are modestly decreased at P30 [11]. Differences are also observed in the fraction of flavins comprised by the parent compound riboflavin. In the *Rho^P23H/+^*, only 3% of total flavins was riboflavin, while riboflavin comprised 25% of total flavins in the *Prph2^R172W^* and 62% in the *Prph2^Y141C/+^.* These differences may be, in part, due to differing rates of degeneration in IRD models; *Rho^P23H/+^* causes much faster retinal degeneration than either of the two *Prph2* mutations, and reduced flavin levels may correlate with reduced rod number.

In contrast, the effect of removing *Rtbdn* on total flavin levels is consistent across the three IRD models. In each case, total retinal flavin levels are reduced at P30 (in the *Prph2^Y141C/+^/Rtbdn^−/−^*, *Rho^P23H/+^/Rtbdn^−/−^*, and *Prph2^R172W^/Rtbdn^−/−^*) compared to WT and are similar to those in the *Rtbdn^−/−^*. There is some variability though in the composition of this total flavin pool across models. In the *Prph2^Y141C/+^/Rtbdn^−/−^* and *Rho^P23H/+^/Rtbdn^−/−^*, riboflavin, FAD, and FMN are all reduced compared to WT and are similar to their levels in *Rtbdn^−/−^* retina. However, in the *Prph2^R172W^/Rtbdn^−/−^*, FAD is reduced compared to WT, but both FMN and riboflavin levels are similar to WT. Interestingly, we did not observe any differences in flavin levels at P15, even in the *Rtbdn^−/−^*. This may be because, at P15 (a time shortly after eye opening), the retina is still developing, in particular with regards to OS length and energy consumption, and, in fact, flavin levels increase as the retina matures from P15 to P30.

The mechanisms by which removal of RTBDN worsens retinal degeneration and the precise function of RTBDN remain unclear. One potential mechanism is that RTBDN promotes concentration or cellular uptake of retinal flavins which are essential for cellular and metabolic homeostasis. This hypothesis is supported by our observation that RTBDN levels are up during degeneration (a condition with increased energy requirements) and the finding that flavin levels are reduced in IRD models in which RTBDN is absent. However, it seems likely that RTBDN has additional functions, as well. For example, total flavin levels are similar in IRD models on the *Rtbdn^−/−^* background and the *Rtbdn^−/−^*, while degeneration is accelerated, suggesting that changes in flavin levels alone cannot account for the degenerative effects of removing RTBDN. One additional function for RTBDN may be simply flavin binding. Flavins are almost always bound to proteins, such as albumin and immunoglobulins in the blood and flavoenzymes inside the cell, and no previous flavin binding proteins have been identified in the interphotoreceptor matrix. Precisely regulated flavin binding and uptake are essential as riboflavin can act as a phototoxic agent, generating reactive oxygen species and cellular toxicity [38,39]. The balance between these phototoxic effects and the native antioxidant effects of flavins are thought to be tied to the properties of the surrounding cellular environment but are incompletely understood [40]. The high light environment in the retina make it a logical place for enhanced riboflavin-associated phototoxicity, and it is possible that RTBDN plays a role in preventing accumulation of riboflavin-associated toxic byproducts. Support for this hypothesis comes from our previous in vitro experiments showing that retbindin was capable of preventing light-induced riboflavin-associated cell death [41].

In conclusion, we here present data showing that removing *Rtbdn* accelerates retinal degeneration in a variety of cone- and rod-dominant retinal degenerations. Our combined findings to date indicate that the effects of removing *Rtbdn* are most severe in slower forms of retinal degeneration. This has implications for the broader goal of developing common therapeutic targets capable of providing benefit across multiple IRDs and suggests that preserving cellular and metabolic homeostasis may be the most beneficial for chronic or late-onset diseases. We look forward to future studies evaluating potential specific functions for RTBDN and further exploration into the role of flavins and flavin-metabolism on retinal degeneration.

## 4. Materials and Methods

### 4.1. Animals

*Rtbdn* knockout (*Rtbdn^−/−^)* generated by our lab and previously characterized [8]. *Prph2* mice carrying the Y141C point mutation knocked into the endogenous *Prph2 locus* (*Prph2^Y141C/Y141C^*) were also generated by our lab and characterized previously [13,42]. Mice carrying the P23H mutation knocked into the endogenous rhodopsin locus (*Rho^P23H/P23H^*) were obtained from Jackson Labs (Bar Harbor, ME, USA, stock # 017628) and have also been previously characterized [25]. *Rho^P23H/P23H^* and *Prph2^Y141C/Y141C^* mice were bred onto the *Rtbdn^−/−^* background to generate *Rho^P23H+/−^/Rtbdn^−/−^* mice and *Prph2^Y141C/+^/Rtbdn^−/−^*. WT, *Rtbdn^−/−^*, *Rho^P23H/+^*, and *Prph2^Y141C/+^* mice were also included as controls. All of the animals were in the C57Bl/6/129V background, and they were free of *rd8* and *rd1* mutations. Mice were raised under cyclic light of 12 h dim light (30–50 lux) and 12 h dark conditions. Males and females were equally included into the analyses. All animal procedures and experiments were approved by University of Houston Institutional Animal Care and Use Committee (IACUC, original animal use protocol approval date, 19 October 2016, renewal protocol approved 28 August 2019). For the sample collection, animals were euthanized using CO_2_ asphyxiation. Retina samples for immunoblot and HPLC analyses were collected by winkling method [43] and immediately frozen in liquid nitrogen and stored at −80 °C until used.

### 4.2. Immunoblotting

Fresh retinas were placed in 1X phosphate buffered saline (PBS) (pH 7.2) containing 1% protease inhibitor (PIN) and incubated on ice for 15 min. For IPM fraction, supernatant was collected and saved. The pellets were re-suspended in 0.1X PBS containing 1% PIN and incubated on ice for 15 min. Samples were agitated with the hand-held motor and pestle tip homogenizer and centrifuged at 50,000× *g* for 30 min. The supernatant (cytosolic fraction) was saved, and the pellets were re-suspended in 1X PBS (pH 7.2) containing 1% Triton X-100 and 1% PIN and homogenized with sonication. Samples were incubated at 4 °C for 1 h and later centrifuged at 16,128× *g* for 5 min. The supernatant was used for membrane fraction. The subcellular fractions were run on reducing SDS-PAGE followed by immunoblotting as described previously [1]. Blots were probed with primary antibodies as indicated in the figures and listed in Table 1. Secondary antibodies conjugated to horseradish peroxidase (HRP) were then used as also listed in Table 1. The blots were imaged using Bio-Rad (Hercules, CA, USA) ChemiDoc™ MP imaging system and quantified densitometrically, using Bio-Rad Image Lab v4.1 software. Quantifications were performed on blots with unsaturated bands.

### 4.3. Immunofluorescence Analysis

Eyes were harvested, fixed in Davidson’s fixative, and embedded in paraffin as described in Reference [1]. Immunolabeling on 10 μm paraffin sections was performed as described previously [1]. Briefly, sections were washed in H_2_O, incubated in 1% NaBH_4_ for 2 min, washed in H_2_O, and incubated in blocking buffer (1× PBS, pH = 7.4, 1% fish gelatin, 2% donkey serum, 0.5% Triton X-100, 50 mg/mL bovine serum albumin) for one hour. Sections were incubated overnight in primary antibodies, washed in 1× PBS and incubated with fluorescent secondary antibodies (Table 1) followed by additional washing in 1× PBS. Sections were incubated in DAPI (Thermo Fisher Scientific, Waltham, MA, USA, 62248) for 15 min at a concentration of 0.1 µg/mL. Slides were mounted with Prolong Gold antifade mountant (P36934, Thermo Fisher Scientific, Waltham, MA, USA), and Airyscan images were captured with 63× objective, while other images were captured with 40× objective using a Zeiss (White Plains, NY, USA) 800 LSM confocal system and processed in Zen 2 lite software.

### 4.4. Electroretinography

Full field ERG was performed on dark-adapted animals at P15, P30, P60, and P90 as described previously [47] using UTAS system (LKC, Gaithersburg, MD, USA). Briefly, after anesthetizing animals, eyes were dilated with 1% cyclopentolate for five min. Subsequently, a single drop of Gonak (2.5% hypromellose) was applied, and platinum wire loop electrodes were placed on the eyes. Scotopic measurements were recorded in response to a single flash of white light at the 157.7 cd s/m^2^. After 5 min light adaptation (29.03 cd/m^2^), photopic ERGs were recorded in response to 25 flashes at 79 cd s/m^2^ intensity. Results obtained from the right and left eyes were averaged and plotted in GraphPad Prism 8.3 software.

### 4.5. Light Histology and Morphometry

Paraffin sections were cut along the superior-inferior plane through optic nerve and stained with hematoxylin (MHS16, Sigma, Burlington, MA, USA) and eosin (HT110116, Sigma, Burlington, MA, USA). The slides were mounted using permount mounting medium (SP15100, Fisher Scientific, Waltham, MA, USA), and light microscopic images presented in the figures were captured using 40X objective from the inferior side, 500 μm away from the optic nerve using a Zeiss Axioskop 50 (Carl Zeiss, White Plains, NY. USA). For quantifications, cells in the outer nuclear cells layer were counted in 100 μm windows at intervals of 200 μm across the superior-inferior plane using ImageJ.

### 4.6. Cone Counts

Whole eyes from P30 WT, *Rtbdn^−/−^*, *Rho^P23H/+^*, and *Rho^P23H/+^/Rtbdn^−/−^* were harvested and fixed in 4% paraformaldehyde overnight. Retinas were isolated from the eye cups, washed with 1xPBS (pH 7.4) and blocked with 0.5% Triton X-100 and 5% bovine serum albumin in 1xPBS at room temperature for 2 h. The retinas were then incubated in PNA (1:500; Thermo Fisher Scientific, Waltham, MA, USA L21409), which diluted in blocking solution, overnight at 4 °C. Retinas were washed extensively in 1x PBS and four cuts were made to facilitate flat mounting (photoreceptor side up) using permount mounting medium (SP15100, Fisher Scientific, Waltham, MA, USA). Whole mounted PNA labeled retinas were imaged with Zeiss 800 LSM confocal system using a 40X objective and processed in Zen 2 lite software (Thornwood, NY, USA). Captured images, enclosing 135 × 135 μm area, were taken 350 µm away from the optic nerve in each quadrant. Cones were counted using ImageJ counter plugin, and the values from each of the four quadrants were averaged to give a value for each eye.

Paraffin sections from P90 WT, *Rtbdn^−/−^*, *Prph2^Y141C/+^*, and *Prph2^Y141C/+^/Rtbdn^−/−^* animals were obtained and dehydrated as described above for morphometric analyses. The slides were boiled in EDTA buffer for 20 min for antigen retrieval and blocked with 0.5% Triton X-100, 5% bovine serum albumin and 2% donkey serum in 1xPBS (pH 7.4). PNA was applied at 1:500 dilution and kept at 4 °C overnight. The sections were mounted and z-stack images were captured 300 μm inferior and superior to the optic nerve with Zeiss 800 LSM confocal system using a 20x objective. Images were processed in Zen 2 lite software (Thornwood, NY, USA), and cones enclosing 300-μm length window were counted manually on the collapsed z-stack images. The numbers counted from superior and inferior sides were averaged to obtain a single value for that mouse.

### 4.7. Transmission Electron Microscopy

Eyes were enucleated and fixed in 2% glutaraldehyde, 2% paraformaldehyde, 100 mM cacodylate, 0.025% CaCl_2_ [11]. The enucleated eyes were embedded in plastic resin and sectioned as previously described [48]. Transmission electron microscopy (TEM) was performed on plastic sections (600−800 Å thickness), which were stained with 2% (*v*/*v*) uranyl acetate and Reynold’s lead citrate. Images were captured with a JEOL 100CX microscope.

### 4.8. Fundus and Fluorescein Angiogram Imaging

Fundus and fluorescein angiogram imaging was performed as described previously [49], using a Phoenix micron IV system (Phoenix Research Laboratories, Pleasanton, CA, USA). Briefly, animals were anesthetized with ketamine/xylazine, and the eye was dilated using 1% cyclopentolate for five min. Subsequently, one drop of Gonak was applied to the cornea, and the Micron IV objective was moved into place for imaging. Brightfield images were captured first, and mice were subsequently injected (IP) with fluorescein (Akorn Ak-fluor 10%) at a dose of 0.02 mL per 10 gm weight of the animal. Thirty s after injection, images were collected using the 451.5–486.5 nm excitation and 488 nm emission green fluorescent protein (GFP) filter. All images were captured using the same light intensity and gain settings using Discover-1.2 software (freeware, San Antonio, TX, USA).

### 4.9. Flavin Quantification by HPLC

After 6 h of fasting, animals were euthanized. Tissues collected for flavin analysis were either used fresh or snap frozen and stored at −80 °C until use. The tissue was homogenized with a motor and pestle in 1xPBS (pH 6.8), centrifuged for 10 min at 1000× *g*, and the supernatant was added to 10% trichlororacetic acid (TCA) for protein precipitation. The supernatant was then centrifuged at 10,000× *g* for 10 min, and the supernatant was filtered and used for analysis in the Waters HPLC system (Waters, Milford, MA, USA) was performed as previously described [37]. Sample extraction was performed under dim red light to minimize the photodegradation of flavins. Flavin amounts were measured by calculating the area under the curve (AUC) using the Breeze 2 software (Waters, Milford, MA, USA). The AUC raw data was converted to pmol per retina for final analysis.

### 4.10. Quantification of ATP Levels in Retina

Cellular ATP levels in fresh retinas were measured using the Abcam luminescent ATP detection assay kit (ab113849) according to the supplier’s protocol as described previously [8]. This experiment was performed under dim red light.

### 4.11. Statistical Analysis

Statistical analyses were performed using one- way ANOVA with Tukey’s post-hoc comparison or two-way ANOVA with Sidak’s post-hoc comparison. For very small sample numbers, Kruskall–Wallis test with Dunn’s post-hoc comparison was used. Data for all experiments were expressed as ± SD. Graphpad Prism 8.3 was used for the quantitative analysis.

### 4.12. Reproducibility

Each set of data has been reproduced the number of times (*n*) described in each figure legend. ‘*n*’ refers to the number of retinas, eyes, or animals.

### 4.13. Data Availability

All data supporting the conclusions of this study are available within the manuscript.

## Figures and Tables

**Figure 1 ijms-21-08083-f001:**
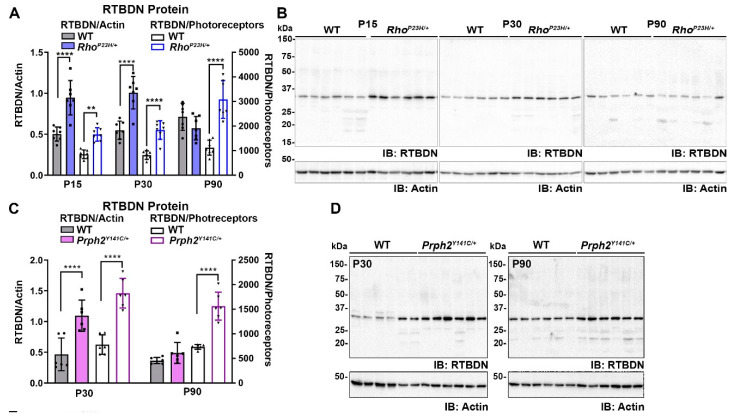
Retbindin (RTBDN) is upregulated in models of retinal degeneration**.** (**A**–**D**) Immunoblot analysis of retinal extracts from indicated genotypes. *n* = 7 different retinas/genotype and three different blots (shown are representatives) for (**B**) and *n* = 6 different retinas/genotype for (**D**). (**A**,**C**) Blots were analyzed densitometrically, and levels of retbindin were normalized to actin (solid bars, left *Y*-axis) or normalized to actin and photoreceptor cell number (white bars, right *Y*-axis). Data presented as mean ± SD. ** *p* < 0.01, **** *p* < 0.0001 by two-way ANOVA with Sidak’s post hoc test. Symbols represent individual retinas. (**E**,**F**) Immunoblot analysis of RTBDN for P30 retinal extracts first separated into soluble inter photoreceptor matrix (IPM), cytoplasm, and membrane fractions. Blots were subsequently probed with anti-IRBP (interphotoreceptor retinoid binding protein, -glyceraldehyde 3-phophate dehydrogenase (GAPDH), and -PRPH2 antibodies serving as controls for each fraction. (**G**,**H**) Representative single-plane confocal images of retinal cross-sections taken from P30 retinas labeled for RTBDN (red), rhodopsin (green), peanut agglutinin (PNA) (teal) in (**G**) or RTBDN (red) and PRPH2 (green) in (**H**). Nuclei were counterstained with 4′,6-diamidino-2-phenylindole (DAPI, blue). RPE, retinal pigment epithelium; OS, outer segment; IS, inner segment; ONL, outer nuclear layer. Magnification, 63X airyscan. Scale bar, 5 μm (**G**), and 2 μm (**H**).

**Figure 2 ijms-21-08083-f002:**
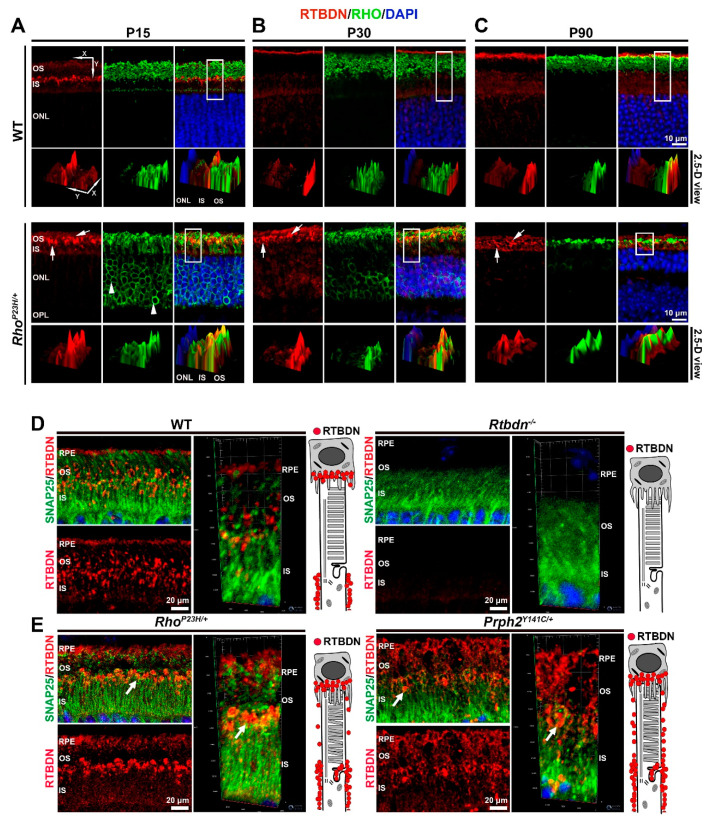
RTBDN exhibits abnormal localization in the *Rho^P23H/+^* retina. (**A**–**C**) Shown are representative single-plane confocal images of P15, P30, and P90 retinal cross-sections taken from WT and *Rho^P23H/+^* mice labeled for RTBDN (red) and rhodopsin (RHO, green), with nuclei counterstained with DAPI (blue). The 2.5-D reconstruction of the area enclosed within the white rectangles is shown below the respective confocal images to assess the spatial distribution and signal intensity of labeling. The x and y coordinates are shown to indicate the clockwise rotation of the image, intensity is shown on the *z*-axis. Magnification, top panel, 40× confocal. White arrows indicate abnormal patchy RTBDN labeling. (**D**,**E**) Shown are representative images from P30 retinas co-labeled for RTBDN (red) and synaptosomal-associated protein, 25kDa (SNAP25, green). Arrows highlight globular RTBDN labeling at the distal tip of the IS. Scale bar: 10 μm (**A**–**C**), 20 µm (**D**,**E**). RPE, retinal pigment epithelium; OS, outer segment; IS, inner segment; ONL, outer nuclear layer, OPL: outer plexiform layer. Experiments were repeated on at least three eyes from three different animals per genotype and age.

**Figure 3 ijms-21-08083-f003:**
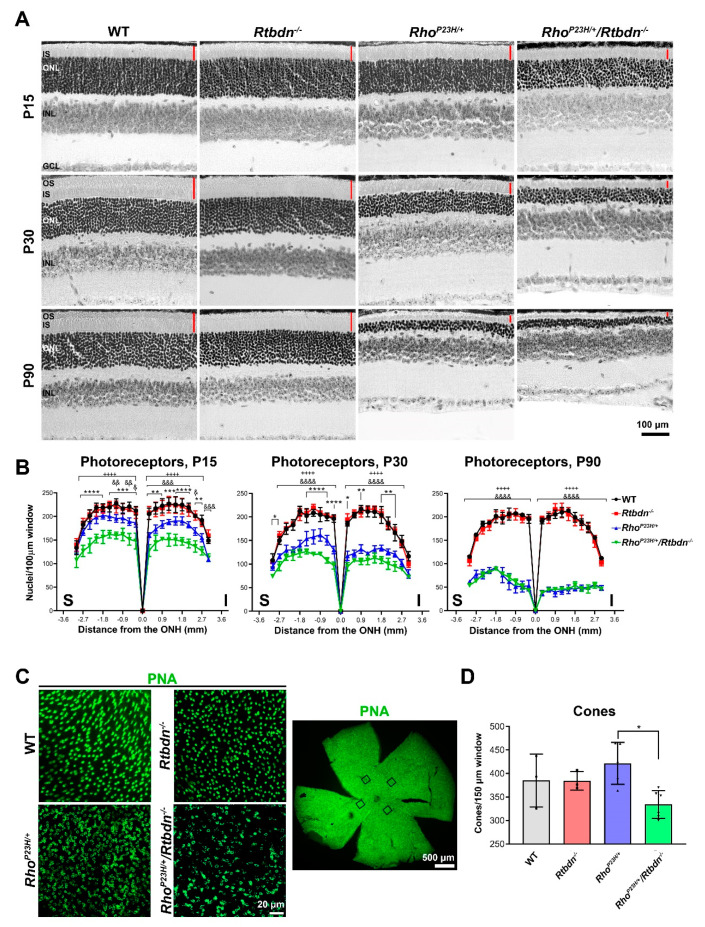
Eliminating *Rtbdn* exacerbates retinal degeneration in the *Rho^P23H/+^* retina. (**A**) Light microscopic images of hematoxylin and eosin (H&E) stained retinal sections from retinas of the indicated genotypes and ages. Red bars highlight the IS/OS layer. (**B**) Photoreceptor nuclei were counted across the superior-inferior plane from retinas at P15, P30, and P90 and plotted in spidergrams (mean ± SD, *n* = 3 eyes/genotype/age). + indicates comparison between wild type (WT) and *Rho^P23H/+^/Rtbdn^−/−^*, & indicates comparison between WT and *Rho^P23H/+^*, and * indicates comparison between *Rho^P23H/+^* and *Rho^P23H/+^/Rtbdn^−/−^.* One symbol, *p* < 0.05, two symbols, *p* < 0.01, three symbols, *p* < 0.001, and four symbols, *p* < 0.0001 by two-way ANOVA with Sidak’s post-hoc test. (**C**) Cone photoreceptors were labeled with PNA (green) in P30 retinal whole mounts from the indicated genotypes. Left side shows example of counted region from the whole mount. Example wholemount highlighting four counted areas is shown on the right. For quantification, 4 independent images were captured at the same distance from the optic nerve head for each retina and (**D**) cone counts for these 4 images were averaged to give a value for each sample. *n* = 3–6 retinas/genotype. Data presented as mean ± SD. * *p* < 0.05 by Kruskall–Wallis test with Dunn’s multiple comparison test. Symbols represent individual animas. RPE, retinal pigment epithelium; OS, outer segment; IS, inner segment; ONL, outer nuclear layer; INL, inner nuclear layer; OPL, outer plexiform layer; GCL, ganglion cell layer; I, inferior; S, superior; ONH, optic nerve head. Magnification, 20×. Scale bar, 100 μm (**A**), 20 μm (**C**, left), and 500 μm (**C**, right).

**Figure 4 ijms-21-08083-f004:**
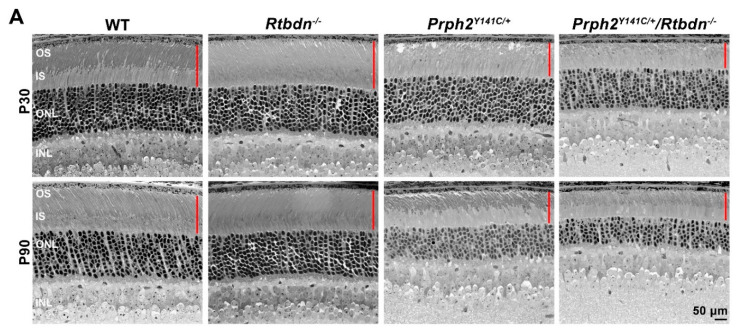
Eliminating *Rtbdn* exacerbates retinal degeneration in the *Prph2^Y141C/+^* retina. (**A**,**C**) Shown are light microscopic images of plastic embedded/toluidine blue stained retinal sections at P30 and P90 (**A**). Red bars indicate OS/IS layer. In (**C**), photoreceptor nuclei were counted across the superior-inferior plane from retinas at P30 and P90 and plotted in spidergrams (mean ± SD, *n* = 3 eyes/genotype). + indicates comparison between WT and *Prph2^Y141C/+^/Rtbdn^−/−^*, & indicates comparison between WT and *Prph2^Y141C/+^*, and * indicates comparison between *Prph2^Y141C/+^* and *Prph2^Y141C/+^/Rtbdn^−/−^.* One symbol, *p* < 0.05, two symbols, *p* < 0.01, three symbols, *p* < 0.001, and four symbols, *p* < 0.0001 by two-way ANOVA with Sidak’s post-hoc test. (**B**,**D**) Cone photoreceptors were labeled with PNA in P30 retinal cross sections. In (**D**), cones were counted from images captured at 300 μm from the optic nerve head for both superior and inferior hemispheres and cone counts for these 2 images were averaged to give a value for each eye. Data presented as mean ± SD, *n* = 3 eyes/genotype, symbols represent individual animals. RPE, retinal pigment epithelium; OS, outer segment; IS, inner segment; ONL, outer nuclear layer; INL, inner nuclear layer; OPL, outer plexiform layer; GCL, ganglion cell layer; I, inferior; S, superior; ONH, optic nerve head. Magnification, 40X (**A**), and 20X (**B**). Scale bar, 50 μm (**A**), and 20 μm (**B**).

**Figure 5 ijms-21-08083-f005:**
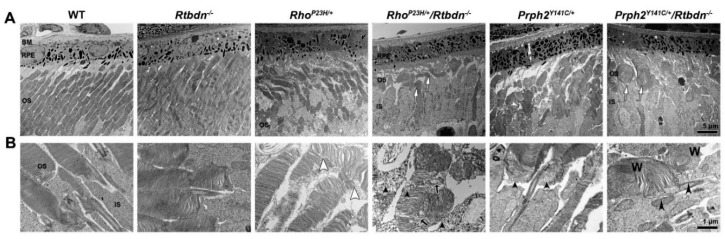
Ultrastructural effect of *Rtbdn* ablation in degenerating retinas. (**A**,**B**) Shown are representative TEM images of the outer retina/RPE (**A**) and the OS (**B**) from the indicated genotypes at P30. White arrows show short, sparse OSs, white arrowheads heads show mis-oriented OS discs, black arrows show unflattened discs, and black arrowheads show abnormal membrane accumulation; “W” indicates whorls. Magnification; 5000× (**A**), 25,000× (**B**). Scale bar 5 μm (**A**), 1 μm (**B**). BM; Bruch’s membrane, RPE; retinal pigment epithelium, OS; outer segment, IS; inner segment. EM images were captured from at least three eyes/genotype.

**Figure 6 ijms-21-08083-f006:**
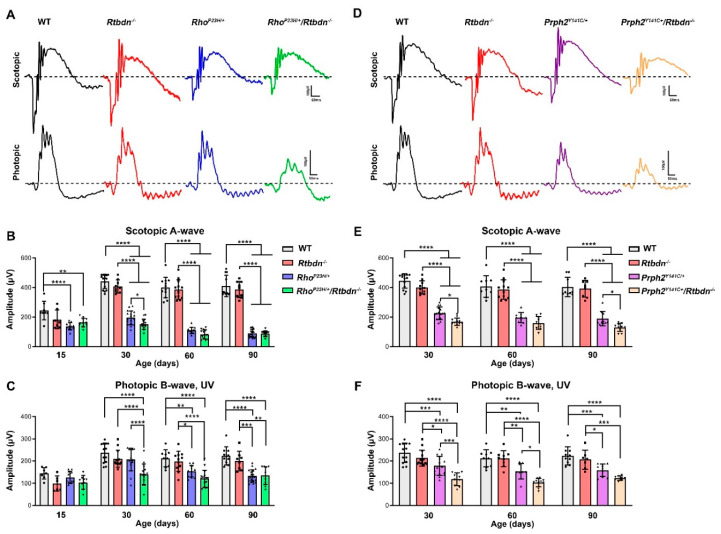
*Rtbdn* elimination exacerbates functional decline in degenerative models. (**A**,**D**) Shown are representative scotopic and photopic waveforms recorded at P30. (**B**,**E**) Maximum scotopic A-wave amplitudes are plotted as mean ± SD from recordings taken at the indicated ages/genotypes. (**C**,**F**) Maximum photopic B-wave amplitudes are plotted as mean ± SD. *n* = 10−19 animals for each genotype/age. * *p* < 0.05, ** *p* < 0.01, *** *p* < 0.001, **** *p* < 0.0001 by two-way ANOVA with Sidak’s post hoc test. Symbols represent individual animals.

**Figure 7 ijms-21-08083-f007:**
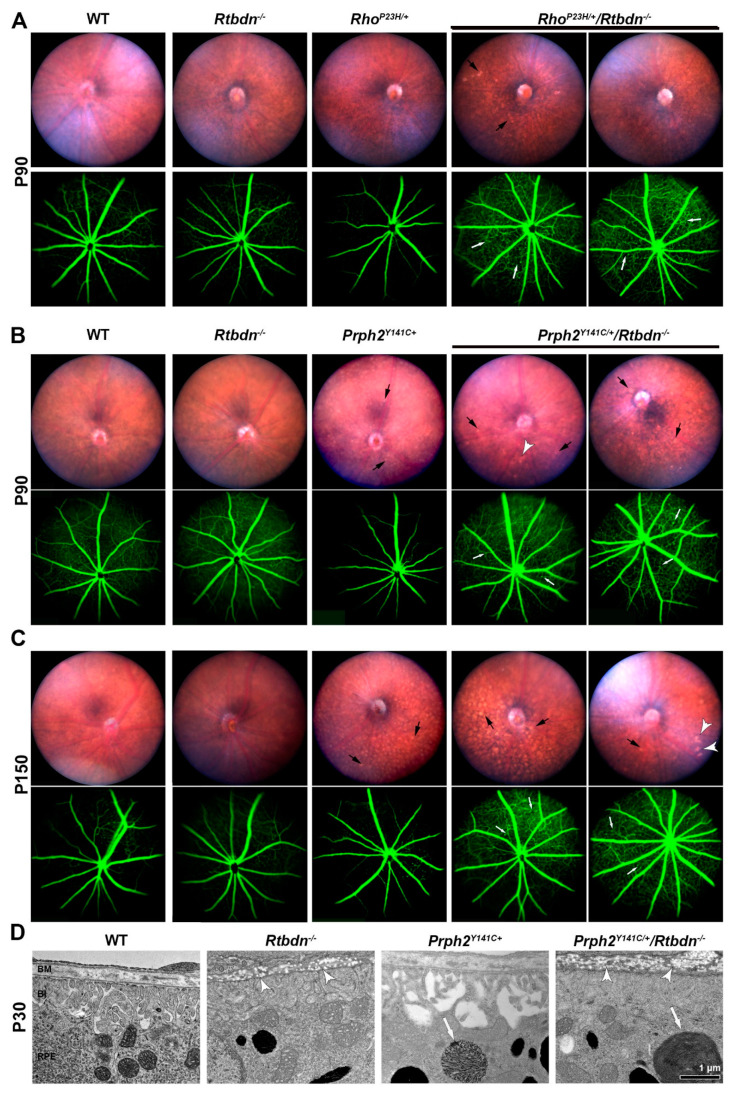
Non-photoreceptor effects of eliminating *Rtbdn* in degenerative models**.** (**A**–**C**) Shown are bright field fundus (top) and fluorescein angiograms (bottom) from mice of indicated genotypes at P90 (**A**,**B**) and P150 (**C**). Two different *Rho^P23H/+^/Rtbdn^−/−^* and *Prph2^Y141C/+^/Rtbdn^−/−^* are displayed for each age. *n* = 8−10 for each genotype/age. White arrow; neovascular tufts, black arrow; mottling or other minor fundus abnormalities, white arrowheads patchy areas of hypopigmentation. (**D**) TEM from P30 eyes of the indicated genotypes showing the RPE and Bruch’s membrane. Arrowheads highlight accumulation of abnormal spots in Bruch’s membrane, arrows highlight large remnants of undigested OS material in the RPE. BM, Bruch’s membrane; RPE, retinal pigment epithelium. Magnification, 25,000×. Scale bar: 1 μm.

**Figure 8 ijms-21-08083-f008:**
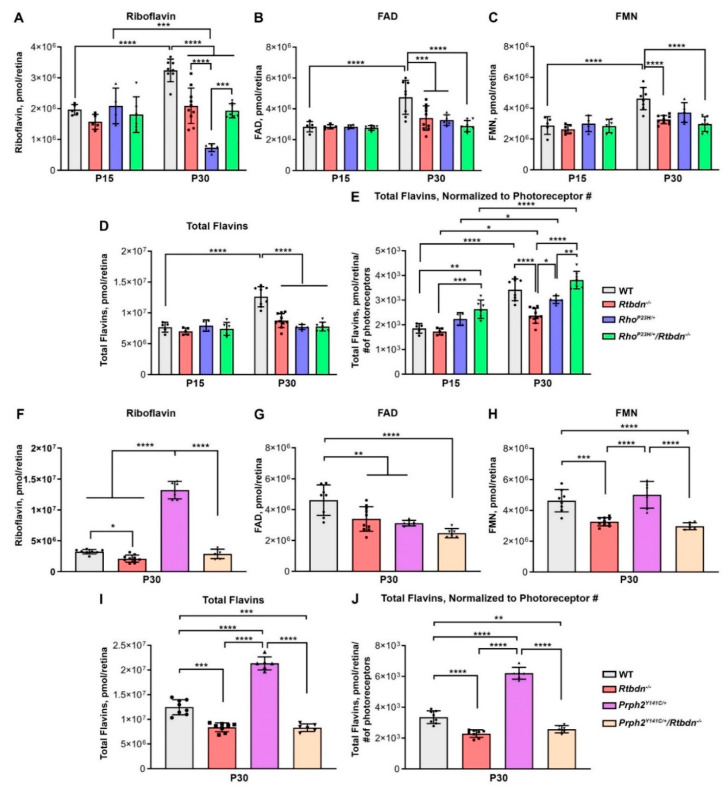
Flavin levels are altered in models of retinal degeneration. Shown are levels of riboflavin (**A**,**F**) and its coenzyme derivatives flavin adenine dinucleotide (FAD) (**B**,**G**), and mononucleotide (FMN) (**C**,**H**) (measured by HPLC) from retinas of the indicated genotypes and ages. (**D**,**I**) Plotted are total flavin levels representing the sum of riboflavin, FAD, and FMN levels. (**E**,**J**) Plotted are total flavin levels normalized to the number of photoreceptors remaining in the retina. *n* = (5−6) retinas for each genotype/age. Data are plotted as ± SD. Symbols represent values from independent retinas * *p* < 0.05, ** *p* < 0.01, *** *p* < 0.001, **** *p* < 0.0001 by two-way ANOVA with Sidak’s post hoc test (**A**–**D**) and one-way ANOVA with Tukey’s post-hoc test (**E**)–(**H**).

**Table 1 ijms-21-08083-t001:** Antibodies used. Antibodies and associated parameters are listed. WB: western blot, IF: immunofluorescence.

Antigen	Species	Clone	Application/Concentration	Source
Retbindin	Rabbit		1:500 (WB,IF)	In-House [1,44]
Prph2	Mouse	2B7	1:1000 (WB, IF)	In house, [45] Available from Millipore Cat# MABN2395
IRBP	Rabbit		1:1000 (WB)	Gift from Dr. Gregory Liou [46]
GAPDH	Mouse		1:1000 (WB)	Abcam Cat# Ab8245, RRID:AB_2107448
Rhodopsin	Mouse	1D4	1:1000 (IF)	Gift from R. Molday Can be purchased: Santa Cruz Biotechnology Cat# sc-57432, RRID:AB_785511
SNAP25	Mouse	SMI-81	1:1000 (WB, IF)	Covance Cat# SMI-81R-100, RRID:AB_510034
Actin-HRP	Mouse	AC-15	1:50,000 (WB)	Sigma-Aldrich Cat# A3854, RRID:AB_262011
Rabbit IgG HRP	Goat		1:25,000 (WB)	Milipore Cat# AP187P, RRID:AB_92625
Mouse IgG HRP	Goat		1:25,000 (WB)	Millipore Cat# AP181P, RRID:AB_11214094
Rabbit IgG Alexa Flour 647	Goat		1:1000 (IF)	Thermo Fisher Scientific Cat# A21245, RRID: AB_2535813
Mouse IgG Alexa Flour 555	Donkey		1:1000 (IF)	Thermo Fisher Scientific Cat# A-31570, RRID:AB_2536180
PNA Alexa Fluor 488			1:500 (IF)	Thermo Fisher Scientific Cat# L21409, RRID:AB_2315178
DAPI (stain)			1:1000 (IF)	Thermo Fisher Scientific Cat# 62248

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
