# Peer review of "Retbindin: A riboflavin Binding Protein, Is Critical for Photoreceptor Homeostasis and Survival in Models of Retinal Degeneration"

_ijms, 2020, doi:10.3390/ijms21218083_

Round 1
Reviewer 1 Report
The manuscript by Genc et al describes the phenotype of two new models of retinal degeneration. The manuscript is well written and organized. The authors here compiled a substantial amount of data.
However, a few minor details need to be addressed before the manuscript can be accepted for publication.
- Figure 3 and 4, The authors should indicate in the spidergram which area is inferior and superior, this because on B (P30 and P90) the right size seems thinner. Also, the authors should clarify what they are compared to be easy for the reader to interpreter the symbols (* and &).
- Figure 4. The ERG data should be presented as box-and-whisker plot, where boxes indicate the 25% and 75% quantile range and whiskers indicate the 5% and 95% quantiles, and the intersection of line and error bars indicates the median of the data. Authors should also increase the size of the scale bar on A and D.
- At P15 the retina is not yet fully developed, therefore performing ERG at this time point is complex.
- Figure 8 is too small, being quite hard to read.
- Material and method section needs to be improved, instead of referring to previous studies, the authors should describe in text what was done.
Author Response
The manuscript by Genc et al describes the phenotype of two new models of retinal degeneration. The manuscript is well written and organized. The authors here compiled a substantial amount of data.
Thank you for these kind comments.
Figure 3 and 4, The authors should indicate in the spidergram which area is inferior and superior, this because on B (P30 and P90) the right size seems thinner. Also, the authors should clarify what they are compared to be easy for the reader to interpreter the symbols (* and &).
On figure 3-4 superior and inferior are indicated by S, and I on the graphs and defined in the legends. We apologize if this was unclear and have made the labels larger so they are easier to see. As indicated in the figure 3 legend, + indicates comparison between WT and RhoP23H/+/Rtbdn-/-, & indicates comparison between WT and RhoP23H/+, and * indicates comparison between RhoP23H/+ and RhoP23H/+/Rtbdn-/-. One symbol, P<0.05, two symbols, P<0.01, three symbols, P<0.001, and four symbols, P<0.0001 by two-way ANOVA with Sidak’s post-hoc test. As indicated in the Figure 4 legend, + indicates comparison between WT and Prph2Y141C/+/Rtbdn-/-, & indicates comparison between WT and Prph2Y141C/+, and * indicates comparison between Prph2Y141C/+ and Prph2Y141C/+/Rtbdn-/-. One symbol, P<0.05, two symbols, P<0.01, three symbols, P<0.001, and four symbols, P<0.0001 by two-way ANOVA with Sidak’s post-hoc test. We have added this information to the main body of the text to make it easier for the reader.
Figure 4. The ERG data should be presented as box-and-whisker plot, where boxes indicate the 25% and 75% quantile range and whiskers indicate the 5% and 95% quantiles, and the intersection of line and error bars indicates the median of the data. Authors should also increase the size of the scale bar on A and D.
In keeping with the request from this reviewer and the second reviewer, we have changed the ERG data (and all other graphed data in the manuscript) to show standard deviations rather than standard errors. Furthermore, for all data except for spidergrams, we have added each individual datapoint to the bar graph so the full range of the data can be appreciated. Adding each individual datapoint to the spidergrams would make the graph very busy, but plotting with standard deviation should give the reviewers a better idea of the consistency in the data.
At P15 the retina is not yet fully developed, therefore performing ERG at this time point is complex.
We agree that P15 ERGs are complex. Although the retina is post-mitotic at this age and the eyes have opened, the outer segments have not fully elaborated. As a result, ERGs in the normal retina increase from P15 to P30. We use P15 ERGs in this study because the degeneration in the RhoP23H/+ model is early-onset and it is valuable to be able to evaluate animals as early in the disease process as possible.
Figure 8 is too small, being quite hard to read.
We apologize for this. We have rearranged Figure 8 and made it larger so it should be easier to read.
Material and method section needs to be improved, instead of referring to previous studies, the authors should describe in text what was done.
Although reviewer 2 indicated that the methods section did not need additional information, for the sake of clarity, we have expanded methods descriptions in several sections.

Reviewer 2 Report
Genc et al. present a study in which they analyzed the impact of the protein RTBDN on various inherited retinal degenerative diseases. They hypothesize that a more general approach to maintain or recover a healthy retinal environment might be a more appropriate treatment approach than corrective therapies for single genes as more than 300 different mutations are responsible for inherited retinal degeneration. RTBDN might be a suitable candidate for such an approach and consequently, the authors tested its impact in retinal degeneration in two different models of rod and cone degeneration.
The authors detected a significant upregulation of RTBDN in both models and revealed that rod and cone degeneration worsened in disease models additionally lacking RTBDN and conclude that the upregulation is a defense mechanism that slows-down the degenerative process. Additionally, this protective mechanism seems to be activated independent from the examined disease models and underlying mutation though the effect is increased in slow-degenerating models.
As reviewed by the authors, the development of treatments for inherited retinal degenerative diseases is, amongst other reasons, hampered by the fact, that multiple mutations cause multiple, mostly rare, forms of IRD. The development of specific therapies for all these mutations, keeping in mind that every day new mutations are identified, is probably impossible. Thus, the hypothesis of recovering a healthy, protective retinal environment as a treatment approach for various diseases seems promising. Consequently, the identification of promising candidates and the development of such therapies is of high impact. The data shown in the present manuscript support the authors’ conclusion of RTBDN as such a potential candidate that may act protective in both, rod and cone dystrophies.
In general, the introduction summarizes well the current knowledge, the hypothesis and gained results. The study is well designed and performed, though the sample size is very small in some analyses (e.g., N=3 for analyses shown in Fig. 3). Fortunately, the variety of analyses showing consistent results that strengthens the conclusion. Nevertheless, in future studies it should be tried to work with bigger groups.
The sketches of the photoreceptors nicely visualize the abnormal localization pattern of RTBDN in fig. 2; in general, data are well presented and explained. Also the methods are described in sufficient detail as many of them were already published elsewhere. The data shown are of high interest and impact for the field offering new avenues for the treatment of IRD.
Major comments
- Page 4, fig. 1 A and B: Comparing the bar graph with the blot, the highly significant difference between WT and KO at P30 is not comprehensible. Please comment on this incoherence.
- The number of samples per group is small (sometimes only 3). Please discuss this limitation.
- The performance of an ANOVA seems inappropriate with such low sample numbers (normal distribution cannot be confirmed and more robust non-parametric tests might be more useful). Please comment on the choice of the statistical test and may correct it.
- It is well discussed why “broad” treatment approaches that aim to recover a healthy retinal environment might be preferred to single-gene corrections and shown data support RTBDN as a potential retinal protective candidate. However, a suggestion/discussion how this knowledge might be translated to a treatment is missing. RTBDN is already upregulated endogenously to slow-down degeneration, so would the authors expect a further increased protective effect if RTBDN is administered (e.g., as recombinant protein or via gene therapy)? Are studies planned to evaluate RTBDN as a therapeutic agent? A discussion of this item in the discussion would be appreciated.
- Referring to comment 6. it has to be mentioned that the protective effect of RTBDN has been analyzed only indirectly (by removing it), but a group in which RTBDN was delivered to the retina is missing. The study is already sophisticated and the addition of this group in this work might be not necessary, but a discussion of the limitation, if a study including this group is planned in future, and how RTBDN could be delivered would be appreciated.
Minor comments
- It seems there are some redundant spaces. Please verify and correct this if necessary.
- The bar graphs always show the SEM instead of the SD, indicating high variances – what might be due to the low sample numbers. Please explain the choice.
- Page 5, legend fig. 1: The first bracket before the “A” should not be written in bold.
- Page 5. Legend fig. 1: “G-H” are not written in brackets.
- Page 17, chapter 4.2 Immunoblotting: Please introduce the abbreviation PIN.
- Page 19, chapter 4.6 cone counts: Please explain how you washed the retinas (“…Retinas were washed extensively…”).
- Page 19, chapter 4.8 Fundus and Fluorescein Angiogram Imaging: Please correct the dose of fluorescein (“… 0.01mL per 10 gm…”) and add a space after 0.01 mL.
Author Response
Genc et al. present a study in which they analyzed the impact of the protein RTBDN on various inherited retinal degenerative diseases.…The data shown in the present manuscript support the authors’ conclusion of RTBDN as such a potential candidate that may act protective in both, rod and cone dystrophies.
Thank you for these kind comments.
The study is well designed and performed, though the sample size is very small in some analyses (e.g., N=3 for analyses shown in Fig. 3). Fortunately, the variety of analyses showing consistent results that strengthens the conclusion. Nevertheless, in future studies it should be tried to work with bigger groups.
Thank you. We do not have additional tissues to include in our analysis at this time, and as the reviewer notes, we have many different assays which support our conclusions. However, in the future we will use larger groups in histological assays (where N=3) as recommended by the reviewer. We routinely use larger groups for functional (ERG) studies (N=10-19 animals/group) and biochemical assays (N=5-6 animals/group).
Major comments
Page 4, fig. 1 A and B: Comparing the bar graph with the blot, the highly significant difference between WT and KO at P30 is not comprehensible. Please comment on this incoherence.
In Figure 1A-B, the increased protein expression at P15 between RhoP23H/+ and WT is easily visualized. At P90, there is decreased expression in RhoP23H/+ compared to WT, due to ongoing degeneration which is also visualized. However, the reviewer is right that it is difficult to see the increased expression at P30 where the bands appear similar. This is both because only 2 samples of each genotype are shown on the blot and because the blot display in the P30 panel is saturated. All quantification is done on unsaturated blots, and we have replaced the saturated blots with unsaturated ones in the revision. In addition, we have replaced all the blots in Fig 1B (showing only 2 samples per group/age) with blots showing 6 independent retinas per group/age so that the reviewers can see more samples.
The number of samples per group is small (sometimes only 3). Please discuss this limitation. The performance of an ANOVA seems inappropriate with such low sample numbers (normal distribution cannot be confirmed and more robust non-parametric tests might be more useful). Please comment on the choice of the statistical test and may correct it. The bar graphs always show the SEM instead of the SD, indicating high variances – what might be due to the low sample numbers. Please explain the choice.
As mentioned above, for cone counts in some cases, we have only three samples, which are not normally distributed, so for these studies we have changed from ANOVA to the Kruskal-Wallis test, which is more appropriate for non-Gaussian distributions. In addition, in keeping with the request from this reviewer and the first reviewer, we have changed all graphed data in the manuscript to show standard deviations rather than standard errors. Furthermore, for all data except for spidergrams, we have added each individual data point to the bar graph so the full range of the data can be appreciated. Adding each individual data point to the spidergrams would make the graph very busy but plotting standard deviation instead of standard error gives a better idea of the consistency of the data. Because the spidergram data are consistent from animal to animal and because we have multiple methods addressing our conclusions (as the reviewer mentioned above) we are confident that the small sample size in this case adequately represents the data. However, in the future we will consider enlarging the sample size.
It is well discussed why “broad” treatment approaches that aim to recover a healthy retinal environment might be preferred to single-gene corrections and shown data support RTBDN as a potential retinal protective candidate. However, a suggestion/discussion how this knowledge might be translated to a treatment is missing. RTBDN is already upregulated endogenously to slow-down degeneration, so would the authors expect a further increased protective effect if RTBDN is administered (e.g., as recombinant protein or via gene therapy)? Are studies planned to evaluate RTBDN as a therapeutic agent? A discussion of this item in the discussion would be appreciated.
Referring to the prior comment. it has to be mentioned that the protective effect of RTBDN has been analyzed only indirectly (by removing it), but a group in which RTBDN was delivered to the retina is missing. The study is already sophisticated and the addition of this group in this work might be not necessary, but a discussion of the limitation, if a study including this group is planned in future, and how RTBDN could be delivered would be appreciated.
Thank you for the thoughtful remarks. The reviewer is correct that our proof of principle studies here on the critical role of RTBDN evaluate the effects of its absence, which makes degeneration worse. This is a critical first step and we are absolutely planning subsequent studies to evaluate the effects of overexpression. We will likely begin with genetic overexpression since using transgenic models to overexpress retbindin would overcome some problems with gene therapies like incomplete transduction, low gene expression levels, etc. However, the ultimate test would be to deliver Retbindin as a genetic therapy, perhaps in nanoparticles as our lab has extensive expertise in that area, and evaluate the effectiveness of retbindin as a therapy as well as the timecourse needed to observe beneficial effects (i.e. prior to onset of disease? Mid-disease?). While these studies are beyond the scope of this work, we are excited to proceed in this direction and have added discussion of this issue to the discussion section of the manuscript.
Minor comments
It seems there are some redundant spaces. Please verify and correct this if necessary.
Page 5, legend fig. 1: The first bracket before the “A” should not be written in bold.
Page 5. Legend fig. 1: “G-H” are not written in brackets.
Page 17, chapter 4.2 Immunoblotting: Please introduce the abbreviation PIN.
Page 19, chapter 4.6 cone counts: Please explain how you washed the retinas (“…Retinas were washed extensively…”).
Page 19, chapter 4.8 Fundus and Fluorescein Angiogram Imaging: Please correct the dose of fluorescein (“… 0.01mL per 10 gm…”) and add a space after 0.01 mL.
Thank you for your thoroughness. We have corrected all of these items except the spacing. The spacing appears off because the text is right and left justified, but if the editorial office needs additional changes, we are happy to make them.

Round 2
Reviewer 2 Report
Dear authors,
the new presentation of the data including the SD and the individual datapoints improves significantly the transparency and comprehensibilty of the data.
In figure 1B the new blots better support the conclusion.
The use of the Kruskal-Wallis test for the smallest groups is appreciated.
I do not see further issues that have to be modified and would appreciate to see the manuscript published.